

# Cover crops improve soil structure and change organic carbon distribution in macroaggregate fractions

Norman Gentsch[1*], Florin Laura Riechers[1*], Jens Boy[1], Dörte Schweneker[2], Ulf Feuerstein[2], Diana Heuermann[3], Georg Guggenberger[1]

[1] Institute of Soil Science, Leibniz Universität Hannover, Hannover, 30419, Germany
[2] Deutsche Saatveredelung AG, Steimker Weg 7, 27330 Asendorf, Germany
[3] Leibniz Institute of Plant Genetics and Crop Plant Research Gatersleben, Corrensstraße 3, 06466 Seeland, Germany

*Correspondence to*: Norman Gentsch (gentsch@ifbk.uni-hnnover.de)

* N. Gentsch and F. L. Riechers contributed equally to this work as first authors.

## Abstract

Soil structure is sensitive to intensive soil management. It can be ameliorated by a reduction in soil cultivation and stimulation of plant and microbial mediators for aggregate formation, latter a prerequisite and measure for soil quality. Cover crops (CC) are part of an integrated approach to stabilize or improve soil quality. Thereby, the incorporation of diverse CC mixtures is hypothesized to increase the positive effects of CC applications. This study entailed an investigation of the legacy effect of CC on soil aggregates after three crop rotations in the second main crop (winter wheat) after the last CC treatment. Four CCs (mustard, phacelia, clover, and oat) cultivated in pure stands and a fallow treatment were compared to a mixture of the four CC species (Mix4) and a highly diverse 12 plant species mixture (Mix12) in a long-term field experiment in Germany. The organic carbon (OC) distribution within macroaggregate fractions (16-8, 8-4, 4-2, 2-1 and < 1 mm) and their aggregate stability were measured by dry and wet sieving methods, and the mean weight diameter (MWD) was calculated from water-stable aggregates.

The results showed that compared to the fallow, all CCs increased the MWD between 10 and 19% in soil under the following main crop. The average MWD increase over the fallow was slightly higher for CC mixtures (16%) than for single CCs (12%). Higher MWD improvement at the 20-30 cm depth also indicates additional benefits from a reduction in the cultivation depth. Structural equation modelling (SEM) suggests that single CCs were more likely to increase OC storage in small macroaggregates < 1 mm, while CC mixtures were more likely to increase OC in the largest fraction (8-16 mm). Different individual CC species or mixtures exhibited a varying involvement in the formation of different aggregate fractions. We provide evidence that litter quality, root morphology and rhizosphere input, which affect microbial mediators of aggregate formation, might be the main reasons for the observed differences between CC treatments. Cover crops are valuable



multifunctional tools for sustainable soil management. Here, we showed that they contribute to structure amelioration in arable soils. Increasing the functional diversity of plant species in CC mixtures could be a strategy to further enhance the positive effects of CC in agroecosystems.

## 1 Introduction

Restoring soil structure is one of the most important tasks in terms of sustainable soil management and adaptation of cropping systems to climate change (Lal, 2015; Williams and Petticrew, 2009; Obalum et al., 2017). In recent decades, soil structure degradation is wide-spreaded and has been observed due to the loss of aggregate stability and changes in aggregate size distribution towards smaller aggregate classes (Boix-Fayos et al., 2001). The causes of soil structure degradation have been attributed to intensification of soil cultivation (Williams and Petticrew, 2009), loss of organic matter (OM) and soil biological

functions (Obalum et al., 2017), unilateral crop rotation and mineral fertilization (Lal, 2009). Aggregate stability can provide important information about soil functioning, which defines soil quality and soil health in agroecosystems (Seybold and Herrick, 2001; Lehmann et al., 2020b). Stable soil aggregates are more stress resistant, decrease soil erodibility, and increase the OM protection capacity (Dungait et al., 2012; Six et al., 2000), and climate resilience of soils (Allen et al., 2011; Six and Paustian, 2014).

Aggregation and the arrangement of the aggregates into a defined pattern, i.e. the soil structure, depends on several parameters and is a product of the interactions between quantity and quality of organic residues entering the soil, indigenous soil OM, mineral constituents, soil organisms, as well as land use history (Tisdall and Oades, 1982). Plants are of particular importance in the management of the soil structure, as the farmer — with the selection of the cultivated plant — can directly intervene in plant-microbial mediated soil processes. There are four pathways through which plants can directly and indirectly influence

aggregate formation: **(1)** Functional root characteristics such as root morphologies vary between plant species and differently affect soil parameters, such as soil pore connectivity, (macro)porosity, or aggregate stability (Kutschera et al., 2009; Bacq-Labreuil et al., 2019; Hudek et al., 2022). **(2)** Litter quality controls the decomposition of root and shoot litter and provides polysaccharides as binding agents for aggregates (Liu et al., 2005). **(3)** Rhizodeposits (root fragments, cell debris, exudates, mucilage) can act as binding materials. **(4)** Plants have a strong impact on soil biota by shaping the microbial community with

their rhizosphere inputs (Reinhold-Hurek et al., 2015). Soil biota is a key factor for aggregate formation, where bacteria and fungi appear to be more important than soil fauna (Lehmann et al., 2017). In this context, the sequence of cultivated plants in crop rotation plays a crucial role in the legacy of microbial processes and OM inputs from litter, roots or microbial residues in arable soils. Therefore, crop rotation and crop history have a strong impact on aggregate formation and stability (Wright and Anderson, 2000; Zhou et al., 2020) and appear to be important tools for soil structure amelioration in arable systems.

Cover crops (CC) are cultivated for purposes of soil degradation protection, nutrient leaching and remediation of soil quality (Thorup-Kristensen et al., 2003; Williams and Petticrew, 2009). The CC biomass is either harvested as fodder for energy production or acts as green manure for the following crop. Previous studies have demonstrated strong positive impacts of CC



on soil structure formation and aggregate stability (Mendes et al., 1999; Dabney et al., 2001; Liu et al., 2005; Blanco-Canqui and Ruis, 2020; Stegarescu et al., 2021). Moreover, root activity was found to be the major driver of higher aggregate stability during CC growth (Stegarescu et al., 2021). A recent study examined functional root traits as a basis for classifying seven different cover crops to improve physical soil properties, including aggregate stability (Hudek et al., 2022). The authors demonstrated that the positive effect of macroporosity and aggregate stability during CC growth depended on the morphology of the individual plant root systems. Four out of seven tested species showed positive trends.

Biodiverse CC mixtures can compensate for the weaknesses of single components and improve the multifunctional positive effects of CC (Couëdel et al., 2019). Cover crop mixtures can have higher root biomass than single components (Heuermann et al., 2019), increase the rhizosphere C input and thereby stimulate microbial biomass and activity (Gentsch et al., 2020; Chavarría et al., 2016). The biogeochemical cycling rate and particularly the fungal community composition appeared to be strongly affected by CC species or mixtures (Cloutier et al., 2020; Thapa et al., 2021). Furthermore, litter from multispecies CC crop mixtures increased the molecular diversity of OM inputs to the soil and the number of substrate niches for microbes (Drost et al., 2020). In concluding the recent research on CC mixtures as compared to single CC cultivars, all four pathways in aggregate formation by plant communities (as outlined above) might have a stronger effect in CC mixtures. However, the question of whether biodiverse CC mixtures are able to increase plant-derived soil structure remediation has not yet been investigated. Further, direct root effects of CCs on soil structure formation during their growth have already been documented (Hudek et al., 2022; Blanco-Canqui and Ruis, 2020) but there is still a lack of information on how long-term these changes are. Here, we refer to the long-term effect that a CC has on the soil beyond its active growing period as the legacy effect.

We hypothesize that CCs alter soil structure according to their root morphology so that CC mixtures which combine diverse functional root traits enhance aggregate stability as compared to single species. We further hypothesize that consecutive cover cropping results in a legacy effect of soil structure improvement for the subsequent main crops. Therefore, we investigated the aggregation pattern of four single and two CC mixtures of four and 12 species, respectively, in comparison to a fallow treatment in a long-term field experiment in Germany. We aimed to explore the legacy effect of CC on soil structure in the second main crop rotation following CC treatments.

## 2 Materials and Methods

### 2.1 Long-term experimental site and soil sampling

The samples were collected from a long-term field trial at the Asendorf field station of the Deutsche Saatgutveredelung AG (DSV), 70 km north of Hanover, Germany (49 m above sea level, 52°45′48.4″N 9°01′24.3″E). The climate is temperate oceanic with an annual mean temperature of 9.3 °C and a mean annual precipitation of 751 mm (long-term mean, 1981–2010). The soil developed from a shallow loess cover over glaciofluvial sand (> 50 cm) and was classified as a Stagnic Cambisol according to the World Reference Base (IUSS Working Group WRB, 2022). Soil texture was a silty loam with low heterogeneity across the field site (Table S1). The soil pH was slightly acidic (pH 6.0–6.4), and soil organic C decreased from



1.6 % in the topsoil (0–30 cm) to 0.8 % in the subsoil (30–60 cm). The experiment was conducted as a fully randomized block design with three field replications per treatment. In total, 21 plots 9 × 9 m in size (including 0.7 m edges) were sampled for this study.

The CC field trials were established in 2015 and incorporated in a conventionally managed 2-year crop rotation with winter wheat (*Triticum aestivum* L.) in the first year, followed by maize (*Zea mays* L.) in the second. The wheat straw remained on

the field and was incorporated into the soil by a cultivator and harrow. The maximum cultivation depth was 15 cm. The maize was harvested as whole plant silage. Seven CC treatments were investigated: (i) fallow with no CC treatment; pure stands as single crops; (ii) mustard (*Sinapis alba* L.); (iii) lacy phacelia (*Phacelia tanacetifolia* BENTH.); (iv) bristle oat (*Avena strigosa* SCHREB), and (v) Egyptian clover (*Trifolium alexandrinum* L.); (vi) Mix4, a mix of the four single species; and (vii) Mix12, a commercial 12-species CC mix (TerraLife® MaizePro TR Greening, DSV, Lippstadt, Germany). Mix12 was 23% legumes

(in shoot dry mass), namely, field pea (*Pisum sativum* L.), crimson clover (*Trifolium incarnatum* L.), alsike clover (*Trifolium hybridum* L.), Persian clover (*Trifolium resupinatum* L.) and Hungarian vetch (*Vicia pannonica* CRANTZ.), and nonlegume species, namely, sorghum (*Sorghum sudanense* STEUD.), common flax (*Linum usitatissimum* L.), lacy phacelia, deeptill radish (*Raphanus sativus* L.), ramtil (*Guizotia abyssinica* CASS.), sunflower (*Helianthus annuus* L.) and camelina (*Camelina sativa* L.). The fallow period was maintained mechanically or by herbicide application.

Soil samples were collected with minimal disturbance on 19 and 20 October 2020, after sowing of winter wheat, which equates to 7 months after incorporating the CC residues. Sampling occurred after completion of the third crop rotation 6 years after the experimental start. The samples were collected from three soil depth increments (0-10, 10-20 and 30-40 cm) by cutting a clod (10x10x10 cm) with a sharp spatula and were carefully transferred to the sample container that fit the size of the clod. Three replicates were collected per treatment and transported (mounted on foam) to the laboratory, air-dried in their clod forms and

gently crushed. After that, the samples were placed in a drying oven at 40°C for 24 hrs. Samples for bulk density (BD) determination were collected by a 100 cm³ stainless core cutter. The cores were dried at 105°C, and BD was determined gravimetrically.

**2.2 Aggregate fractionation and stability index**

Macroaggregate stability was determined according to Hartge and Horn (2009). Dry soil samples from the seven CC variants were separated into five different aggregate sizes by dry sieving. To determine the distribution of the aggregate classes 16-8, 8-4, 4-2, 2-1 and < 1 mm, a nest of five sieves with mesh sizes of 16, 8, 4, 2, and 1 mm (VWR International, ISO 3310, 200x50 mm, stainless steel) was used for manual sieving. From each sample, 120 g of the sieved fractions was transferred to glass backers.

To determine the aggregate stability, the 16-8, 8-4, 4-2 and 2-1 mm aggregate classes from dry sieving were mixed together for wet sieving in proportion corresponding to their original aggregate distribution in the sample. A wet sieving apparatus was used for this purpose according to Hartge and Horn (2009). Each remixed sample was rewetted to 120 % of its dry weight and





placed on the top of a sieve nest with mesh sizes of 8, 4, 2, 1, 0.5 and 0.25 mm. The nest was placed in a holder and suspended in a container of water. Thereafter, the nest was lowered to the point where the soil sample on the top sieve was just covered

with water. The sieves were lowered and lifted by an electric motor with a standardized oscillation of 4.0 cm at a frequency of 38 cycles min$^{-1}$. This procedure was performed for five minutes. After wet sieving, each aggregate class was carefully transferred from the sieves into a glass container. The water from the wet sieving apparatus was decanted after the particles settled. Accordingly, the aggregate classes were flushed again with water into aluminium bowls. After drying at 105°C, the aggregate classes were weighed, and two different stability indices were calculated. The MWD is the mean-weight diameter

of wet-sieved aggregates calculated according to Eq. (1):

$$MWD \ (mm) = \sum_{i=1}^{n} (X_i \times W_i)$$

where $X_i$ is the average diameter in mm of a certain size fraction, $W_i$ is the proportion by weight of aggregates in the size fraction and $n$ is the total number of size fractions (Obalum et al., 2019). The higher the MWD is, the more water stable large

aggregates remain in the sample. The GMD is the geometric mean diameter of wet-sieved aggregates calculated according to Eq. (2):

$$GMD \ (mm) = exp \left( \frac{\sum_{i=1}^{n} Wi \lg X_i}{\sum_{i=1}^{n} Wi} \right)$$

where $W_i$ is the proportion of each aggregate class in relation to the weight of the soil samples (Zhou et al., 2020). The GMD is an estimate of the size of the most frequent aggregate size classes. The higher the value, the larger the mean aggregate size. Both indices assume that larger aggregates imply greater stability (Nimmo and Perkins, 2002).

### 2.3 Soil organic carbon analysis

An aliquot from each aggregate fraction and the bulk soil was homogenized in a ball mill and analysed for organic carbon

(OC), total nitrogen (TN) and stable isotope ratios δ$^{13}$C and δ$^{15}$N using an Elementar IsoPrime 100 IRMS (IsoPrime Ltd., Cheadle Hulme, UK) coupled to an Elementar Vario MICRO cube EA C/N analyser (Elementar Analysensysteme GmbH, Hanau, Germany).

### 2.4 Statistical analyses

Histograms and density plots were used to explore the data distribution. Data were log transformed for statistical tests if the

assumption of normality was not fulfilled. Pairwise t tests (pairwise CI package) were used to compare differences between treatments in R version 4.1.3 (R Core Team, 2022). The overall effect of all CC treatments on MWD in comparison to the fallow was analysed by linear mixed models (LMM, lme4 package) taking CC variant (all treatments) or CC type (single





species vs. mixtures) and soil depth as random variables with varying intercepts among CC and depth. Mean values are provided ± standard error (SE). The impact of CC on OC distribution in aggregate fractions was analysed in a structural
equation model (SEM) using the R package lavaan (Rosseel, 2012). We used a two-step approach for SEM construction. First, a base model was tested to confirm the relationship between the latent variables (constructed) and indicator variables (measured). The base model used a covariance matrix based on the correlation pattern between forcing variables (Fig. S9). The latent variable "aggregate OC distribution" was composed of OC1 (OC in the <1 mm fraction), OC4_2 (OC in the 4-2 mm fraction), OC8_4 (OC in the 8-4 mm fraction) and OC16_8 (OC in the 16-8 mm fraction). Principal component analyses
confirmed a similar loading of OC2_1 and OC4_2 on the first two components (eigenvalue >1), explaining 61% of the variance in the data. As OC2_1 and OC4_2 are redundant variables, including both does not fit to the model structure. Thus, we excluded OC2_1 from the latent variable construction. The second latent variable was "soil properties", which was composed of the OC content (bulk soil), BD and clay content. All global and local fit parameters of the base model indicated that the model fit well to the data matrix (see S2, R markdown file). In the second step, the structural equation was included in the base model. The
aggregate OC distribution was used as an endogenous variable (variable explained from the model) that was predicted by CC type, soil properties and MWD (predictor variables). The global and local fit parameters of the final model fitted the data properly, and all predictor variables showed a significant impact on the aggregate OC distribution. The r-squared values of the variables indicate how much of their variance was explained by the SEM. All statistical statements, models and average values can be recalculated from the metadata and R scripts provided. All data and R scripts for evaluation were uploaded to Zenodo
and are publicly available (DOI: 10.5281/zenodo.7147566). The R markdown file is attached as supplementary material S2.)

## 3 Results

Soil OC concentrations and stocks in the topsoil increased significantly from the start of the experiment in 2015 to 2020 (Figs. S3 and S4). The average OC concentrations increased from $1.80 \pm 0.06\%$ to $2.07 \pm 0.06\%$ in the 0-10 cm layer and from $1.79 \pm 0.06$ to $1.84 \pm 0.06$ in the 20-30 cm layer regardless of CC treatment (Fig. S4). No significant differences in OC stocks
between the CC treatments and fallow were found in 2020. Total OC stocks to 40 cm soil depth ranged from 78.5 to 120.7 t ha$^{-1}$ with no significant differences between treatments (Fig. S5b). Only in the 0-10 cm layer phacelia showed higher OC stocks compared to the other treatments (Fig. S5a). There was a strong negative correlation between OC and BD ($R^2 = 0.44$, $p < 0.001$) but not with soil texture. Reasons for the OC increase in all treatments, including fallow status, are discussed in a supplementary section (S1) but will not be the topic of further discussion here.




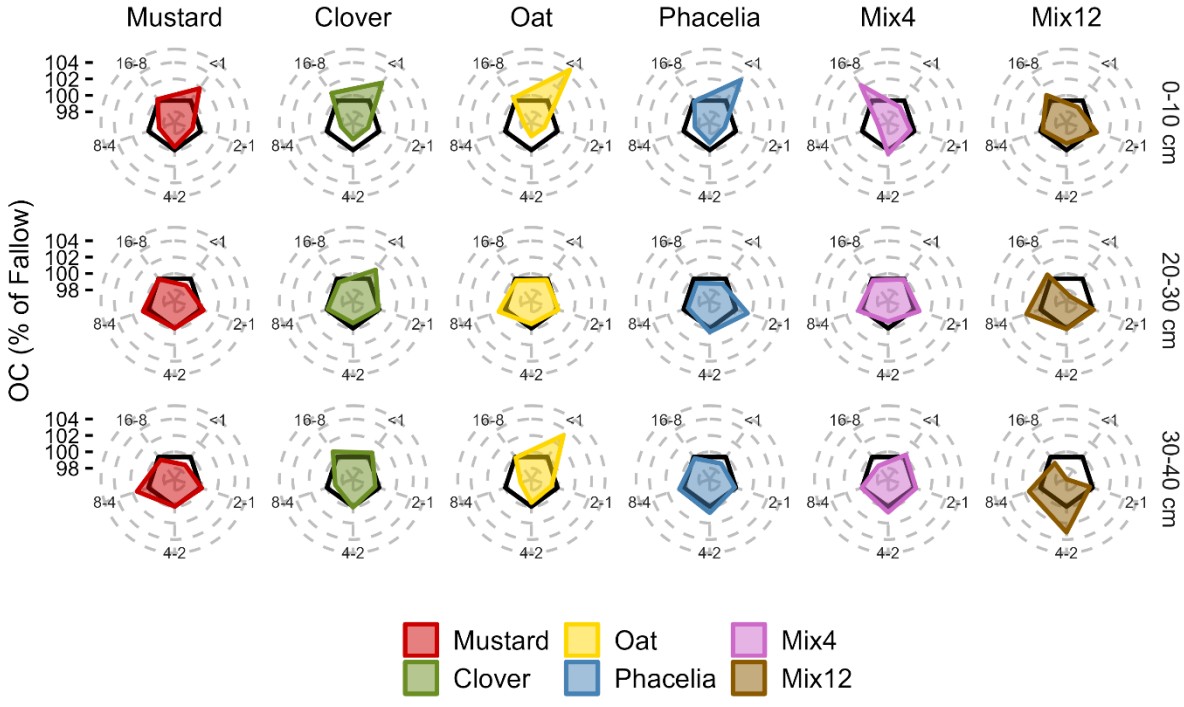

**Figure 1.** Relative proportion of OC in different soil fractions relative to the fallow level. Fallow as 100% is marked by the black polygon in the background. Polygons represent the means of three replicates, and statistical evaluation of the graph is shown in Fig. S8.

Most of the soil aggregates (57.5 to 74.2%) were stored in the < 1 mm fraction, highest in the topsoil and decreasing with soil depth (Fig. S6). The intermediate fractions (8-4 mm, 4-2 mm, 2-1 mm) ranged from 5.6 to 13.8%, each. The largest fraction (16-8 mm) contributed least to the total soil aggregates (0 to 7.0%). A similar distribution was found for the proportion of OC in different aggregate fractions (Fig. S7). The variability of OC distribution within aggregates was quite large between (Fig. 1) but also within CC treatments (Fig. S8). Thus, only a few of the observed trends were significantly different by pairwise comparison of treatments (Fig. S8). For example, the largest differences among CC treatments were found in the top 0-10 cm, between oat and Mix12, Mix4, and fallow. Of all treatments, oat lead to the highest proportion of OC in the <1 mm fraction (Fig. 1), which was a significant difference from Mix12 in the uppermost layer (Fig. S8). All pure cultivated CCs tended to store more OC in the <1 mm fraction in the 0-10 cm increment, whereas mixtures stored a greater proportion in the 16-8 mm fraction. In the soil increments below 20 cm, the data variability was large between treatments. Here, only Mix4 led to approximately 1% more OC in the 2-1 mm fraction than oat at the 30-40 cm depth (Fig. S8).

The MWD varied among treatments and soil depths (Fig. 2a). A larger MWD indicates that more large-scale aggregates are present after wet-sieving. When focusing on individual soil horizons, pairwise comparison indicated significantly higher MWD compared to the fallow for clover at 0-10 cm (18.8% higher), Mix12 at 20-30 cm (37.6% higher) as well as phacelia (17.0%



higher) and Mix4 (12.8% higher) at 30-40 cm. A comprehensive data evaluation of the LMMs indicated that the MWD increased with soil depth and was significantly higher in the CC treatments than in the fallow treatment (Fig. 2b). The LMMs across soil depths indicated that all CCs increased the MWD from 10% for oat and up to 19% for Mix12 compared to fallow

(Table 1, Model 1). Only the increase from oat was not significant at the p level of 0.05. A second model on CC type (Table 1, Model 2) indicated that CC mixtures showed a higher MWD increase (16%) than single species (12%). We explored the relationship between MWD and the soil parameters OC content, texture and BD. None of the parameters showed a significant relationship to MWD (see S2, R markdown file). The CC litter C:N ratio was significantly negatively related to the MWD in the topsoil ($R^2 = 0.35$, p = 0.01, Fig. S9) but not in the subsoil (data on CC C:N ratios were published in Gentsch et al., 2022).

Similar trends were also found with GMD (see S2, R markdown file).

Furthermore, we tested the relationship between MWD and OC in different aggregate size fractions (Fig. S9). There was a significant positive correlation between MWD and OC in the 8-4 mm fraction ($R^2 = 0.25$, p < 0.001), a slightly positive but significant trend with 16-8 mm-sized aggregates (Fig. S9) and a negative correlation with OC in the <1 mm fraction ($R^2 = 0.19$, p < 0.001). Similar trends as we did identify for the MWD were also observed for the GMD, but the evaluation with an

LMM showed significantly higher GMD compared to fallow only for mixed CC (Fig. S10).

The impact of soil parameters and CC type on aggregate OC distribution was evaluated by SEM (Fig. 3). The overall fit parameters indicated that the model fit the data satisfactorily ($\chi^2 = 28.7$, $p = 0.12$, RMSEA = 0.08). The r-square of the SEM variables indicated that 70% of the variance in aggregate OC distribution is explained by the model (see S2, R markdown file). The regression parameters in the SEM indicated that all of the selected predictor variables showed a significant impact on

aggregate OC distribution (p < 0.02). The standardized estimates of the predictors indicated that MWD had the highest impact on aggregate OC distribution, followed by soil parameters and CC type (Fig. 3). The factor CC type was composed of three factors in the order Mix, Fallow, and Single. The interpretation of the CC type in SEM is visualized in the R markdown file (supplementary material S2). By the change in the CC type from Mix to Fallow to Single CCs, the OC distribution is affected positively, which means that OC in the <1 mm fraction increases, while OC in the larger fractions decreases.





**Table 1:** Results of two LMMs with MWD as the response and CC variant or CC type as the predictor variable. Soil depth was set as a random variable. Interpretation: The estimated mean of fallow is the reference group, and changing the treatment from fallow to mustard will increase the MWD by 0.33 mm, which represents an increase of 12.4%.

| Coefficient | Estimate MWD (mm) | Std. Error | df | t value | p value | Sig. |
|---|---|---|---|---|---|---|
| Model 1 CC variant | | | | | | |
| Mean Fallow (intercept) | 2.65 | 0.18 | 4.27 | 14.47 | 0.000087 | *** |
| Fallow - Mustard | 0.33 | 0.16 | 54 | 2.07 | 0.043316 | * |
| Fallow - Clover | 0.34 | 0.16 | 54 | 2.18 | 0.033884 | * |
| Fallow - Oat | 0.28 | 0.16 | 54 | 1.74 | 0.086755 | . |
| Fallow - Phacelia | 0.32 | 0.16 | 54 | 2.05 | 0.045397 | * |
| Fallow - Mix4 | 0.34 | 0.16 | 54 | 2.18 | 0.033698 | * |
| Fallow - Mix12 | 0.51 | 0.16 | 54 | 3.26 | 0.001928 | * |
| Model 2 CC type | | | | | | |
| Mean Fallow (intercept) | 2.65 | 0.18 | 4.15 | 14.57 | 0.000101 | *** |
| Fallow - Mix | 0.43 | 0.13 | 58 | 3.21 | 0.002138 | ** |
| Fallow - Single | 0.32 | 0.12 | 58 | 2.60 | 0.011753 | * |

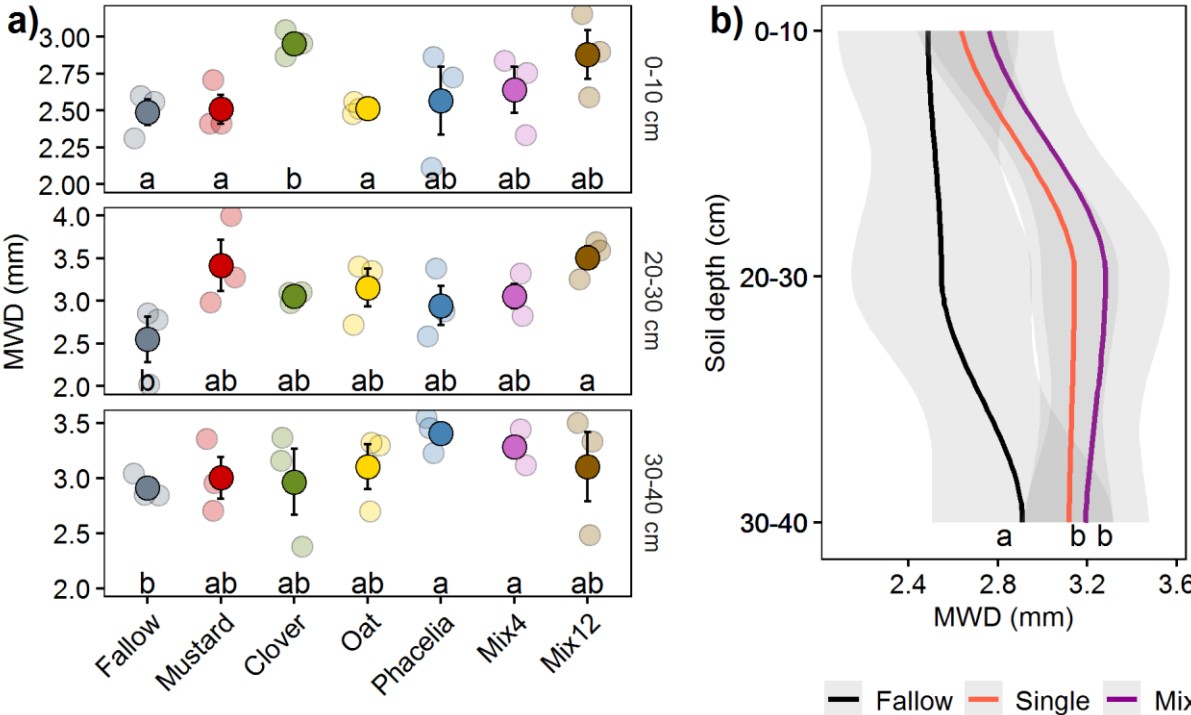

**Figure 2:** Mean weight diameter (MWD of soil aggregates after wet sieving from different soil depths. Lowercase letters denote (a) significant differences between CC treatments by pairwise comparison and (b) overall effects of CC from a LMM. Transparent points represent the individual measurements, and full colours are mean values (±SE). Single = pure cultivated CCs, Mix = CC mixtures.



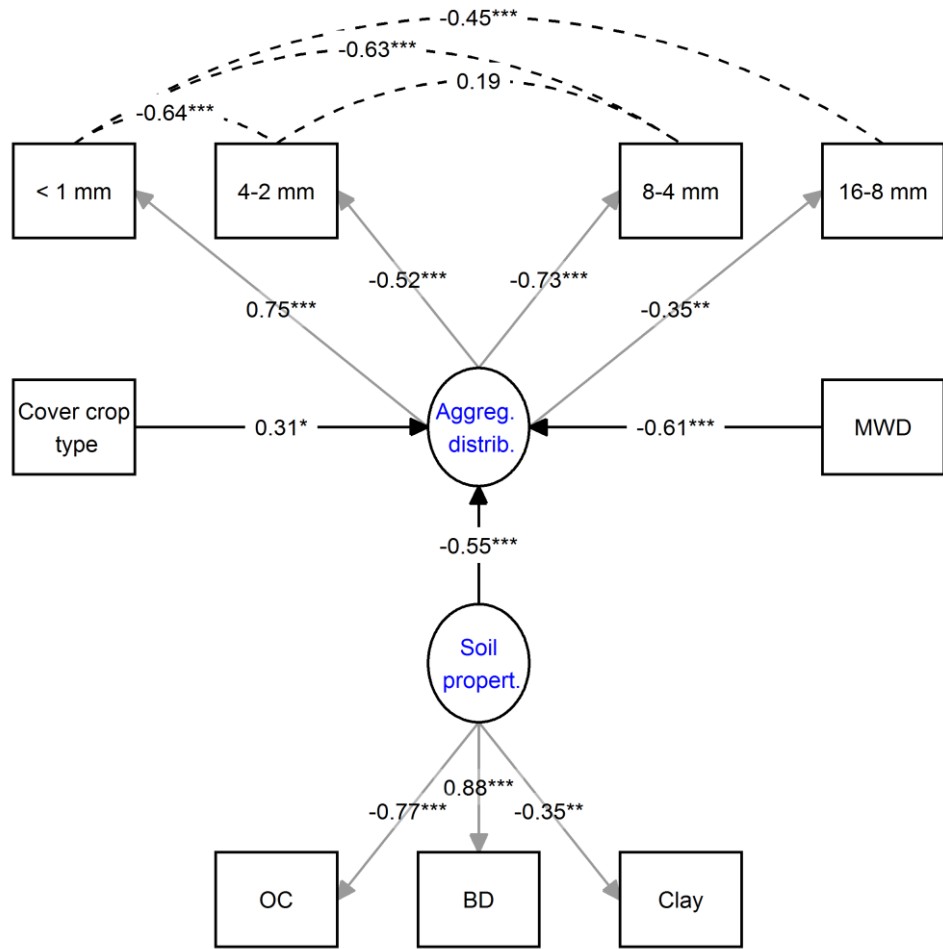

**Figure 3:** Structural equation model (SEM) investigating the impact of parameters on aggregate OC distribution. Latent variables (blue) are predicted by grey arrowed observed variables. Dashed lines indicate covariance variables. Numbers show standardized estimates with p values as asterisks. All model parameters are shown in the R markdown file (supplementary material). MWD = mean weight diameter, BD = bulk density.

## 4 Discussion

Cover crops have substantial impacts on soil properties, but so far it is not clear if a legacy effect of cover cropping can be measured in a second following main crop. Therefore, we analysed the soil aggregate stability that resulted from the consecutive integration of CC in a crop rotation under a winter wheat crop which followed maize 7 months after incorporation





of CC residues. Knowledge on such long-term effects of catch cropping is of great importance to optimize the use of CCs in improving soil properties.

**4.1 Changes in aggregate stability**

The MWD and the GMD both serve as aggregate stability indices assuming that more of the larger aggregates resist wet sieving treatment. The results indicated that the repeated application of CC (Table 1, Fig. 2b) increased the average size of soil aggregates and their stress resistance compared to fallow. Larger aggregates consist of subunits of smaller aggregates that are internally bound stronger to each other as the MWD and GMD indices increase (Nimmo and Perkins, 2002). We found no

correlation between the MWD and soil texture, BD or bulk soil OC content (supplementary material, S2). This suggested that factors other than soil properties determine differences in aggregate stability at the experimental site. The average MWD increase over fallow was slightly higher for CC mixtures (16%) than for single CC (12%). Despite a partially large variability of the data (Fig. 2a), some CC species appeared to have a stronger effect at a specific soil depth than others. Clover significantly increased the MWD in the 0-10 cm increment, while phacelia increased the MWD in the 30-40 cm increment compared to the

fallow (Fig. 2a). As both species are part of the mixtures, these trends can also be observed in the mixtures. Clover, however, was usually only a minor component in Mix4 (<1-5% of the total root biomass, Heuermann et al., 2019), and the effect in 0-10 cm was not present in Mix4. Differences in root morphology between species at the sampling site are well documented in Heuermann et al. (2019) and might be one factor for differences in MWD between species. Clover has its maximum root biomass in the upper 20 cm soil depth, while phacelia can root approximately 70 cm and deeper. In a recent study, total root

length and root surface area correlated positively with aggregate stability, and it was shown that several different CCs led to higher aggregate stability in the topsoil (0-15 cm, Hudek et al., 2022).

Our models indicated that the largest differences in MWD (Fig. 2b) and particularly in GMD (Fig. S10b) between CC and the fallow occurred in the 20-30 cm soil layer. As the maximum cultivation depth during the experiment was 15-20 cm, this indicate that the layers below the cultivation depth benefit most from CC treatments. Soil cultivation and tillage have a strong

impact on aggregate stability and soil structure, resulting in lower MWD (Obalum et al., 2019) and lower connectivity of the pore network (Lucas et al., 2019). Plant activity, on the other hand, has been demonstrated to establish stable and connected biopore systems within 6 years (Lucas et al., 2019). Here, we demonstrate that 6 years after the last ploughing to 30 cm depth (see S1), the incorporation of CC resulted in a faster improvement of MWD and GMD in comparison to fallow treatments. Reduced soil cultivation can be, therefore, even more beneficial for soil structure amelioration if CCs are incorporated into the

crop rotation.

The significantly higher MWD and GMD values of Mix12 at 20-30 cm might indicate additional benefits for aggregate stability when more biodiverse CC mixtures are grown. Apart from root morphology, the quality and quantity of root exudates and plant litter (root and shoot) can be quite different between CC plant species. The metabolite profiling of the four single CCs at the sampling site revealed that every plant species showed a characteristic pattern of chemical compounds that are released





into the soil (Heuermann et al., 2023). In particular, secondary metabolites with signalling functions were closely related to certain plant species. For example, the presence of phenylpropanoids which regulate interactions between plants and their microbial associations, was largely confined to clover (Heuermann et al., 2023). The authors found various plant-specific secondary metabolites with functions for improving iron availability, as chemotactic agents for microbes, as plant-microbial signalling, as signals in plant–plant communication, as allelochemicals, biotic defence or as nitrogen sources for neighbours.

Also the total carbon in field root exudates differed strongly between CCs and was highest for phacelia.

Plants shape the microbial community in their rhizosphere by the composition of their root exudates (Reinhold-Hurek et al., 2015; Ulcuango et al., 2021). Root biomass and root exudates are linked to higher fungal and bacterial biomass (Eisenhauer et al., 2017). Therefore, we suggest that the significantly higher impact of clover in the 0-10 cm layer was derived from the higher rhizosphere input and attraction of microbial mediators suitable for aggregate formation, such as fungi (Lehmann et al., 2020a).

This might also explain the better performance of CC mixtures for aggregate formation compared to single species and the differences between CC treatments in the subsoil. Previous experiments at the same site showed higher photoassimilate C transport rates to the rhizosphere for CC mixtures, prolonged mean residence time of these compounds in the soil and the connection to higher fungal activity compared to single CC (Gentsch et al., 2020). Additionally, Baumert et al. (2018) found significant promotion of particularly fungi by greater exudate release, which are having the largest impact for macroaggregate

formation in the subsoil.

The OM chemistry that is incorporated into the soil plays an important role in the aggregation process. In a study of different OM types, Sarker et al., (2018) used $^{13}$C NMR spectroscopy to evaluate the impact on MWD (referred to as the aggregation index in the study). Materials with high protein contents and low C:N ratios, such as alfalfa litter, showed a rapid positive response to MWD. The study also showed that materials such as maize litter, rich in cellulose and hemicellulose, with wide

C:N ratios but poor in soluble fractions, had an intermediate but persistent response to MWD. In our study, we found a significant negative correlation between MWD and the C:N ratio of CC litter in the 0-10 cm increment but not in the deeper layers (Fig. S9). This suggested that the litter quality of CC residues plays an important role in aggregate stability when incorporated into the topsoil. The C:N ratios were ranked in the order clover, Mix12, phacelia, mustard, Mix4, and oat, suggesting that the positive impact of CC on MWD in the topsoil decreased from clover to oat.

Below the cultivation depth, factors other than litter quality, such as root exudates, might be more prominent for aggregate formation. Fresh plant residues induce the formation of macroaggregates as a result of higher microbial activity (Six et al., 2000). Litter quality exerts control on the decomposer community composition, which will change as a result of decomposition (Marschner et al., 2011; Berg and McClaugherty, 2014). These processes result in microbial-derived organic substances that are key to building up soil aggregates.

We conclude that the combination of plants with different litter qualities and rhizodeposits might explain the better performance of CC mixtures as well as differences between CC species for soil aggregate amelioration. Consecutive integration of CC in the crop rotation resulted in a positive legacy effect on aggregate stability. Reduced cultivation depth in combination with CC mixtures might be an additional benefit to improve the soil structure in the subsoil.



## 4.2. OC distribution in aggregates

The impact of various parameters on aggregate OC distribution was explored by SEM. The model indicates that the MWD and soil properties had the strongest direct effect on aggregate OC distribution (Fig. 3). Soil properties are well known for their contribution to soil aggregate formation. Schweizer et al. (2019), for example, showed the dependency of macroaggregate formation on clay content, while Blanco-Canqui and Lal (2004) emphasized the role of OM quality and quantity in aggregate formation and OC sequestration. A lower but significant effect on the OC distribution was found for the CC type (Fig. 3),

suggesting that it matters if we grow CC instead of fallow as mixtures or single species. Although difficult to detect in pairwise comparisons, mixtures tended to increase OC in the 16-8 mm fraction, while single species supported OC in the < 1 mm fraction, at least in the topsoil (Figs. 1, S7, S8). The SEM supports these trends and suggests that with the change in the treatments from CC mixtures to fallow to single CC, OC in the <1 mm fraction will increase, while OC in the larger fractions will decrease. This relationship might indicate that root performance factors, such as root morphology, rooting depth, and

rhizosphere input of CC contribute to the different OC distributions in aggregate fractions. The tendency of more OC in larger soil fractions might be one of the reasons for the increasing MWD after mixed CC treatments and the positive linear relationship between MWD and OC content in fractions 4-8 and 8-16 mm (Fig. S9). These correlations also underscore that OC stored in the two largest fractions is more stable to withstand induced forces from our wet sieving treatment.

Management practices were found to have a strong impact on OC distribution within aggregates, and a reduction in tillage

resulted in higher OC storage in macroaggregates compared to conventionally tilled soils (Blanco-Canqui and Lal, 2004). Here, we demonstrated that crop history has a legacy effect on the OC distribution and that the incorporation of CC can change the OC distribution within aggregate classes. Classical concepts (such as Tisdall and Oades, 1982) attribute plants and crop management to strong factors for macroaggregate formation, where the number of stable macroaggregates decreases when rhizosphere products and hyphae are decomposed and not replaced. Later studies confirmed the role of crop rotation in the

management of aggregate binding agents (Chan and Heenan, 1999; Blanco-Canqui and Lal, 2004). Overall, our results demonstrate that CC incorporation attenuate negative effects on soil structure that come from soil cultivation. The compensation effect was even stronger when biodiverse CC mixtures were applied.

## 5 Conclusions

This study demonstrates that the stability of macroaggregates and OC distribution in aggregate fractions in arable soils partially

depend on crop rotation and incorporation of CCs. We argue that CC aggregate formation depends on the litter quality, root morphology and rhizosphere input of CCs, which stimulate microbial mediators for aggregate formation. Litter quality had positive effects in the topsoil (0-10 cm) when CC residues with low C:N ratios were incorporated into the topsoil. Layers below the cultivation depth (> 15 cm) showed the strongest response to CC treatments, and root morphology together with exudate-stimulated microbial growth was more likely the factor for the positive effects observed. Cover crop mixtures tended

to have additional benefits for the MWD over single CC and increased OC accumulation in large aggregate fractions (16-8

mm). The consecutive integration of CC in crop rotation can be used to ameliorate aggregate stability in cultivated soils. Reduced soil cultivation depth can be, even more beneficial for soil structure amelioration if CCs are incorporated into the crop rotation.

Stable and agglomerated aggregates are core parameters for soil quality and one of the key indicators in the soil health concept (Lehmann et al., 2020b). Aggregation affects root penetration, decreases the risk of soil erosion and increases hydraulic functions. Aggregates are hotspots for biogeochemical cycling (Costantini and Mocali, 2022) and contributes to plant nutrition and higher crop yields (Li et al., 2021). Therefore, we conclude that the improvement of soil structure we observed, is one of the factors mediated by CC, leading to higher and resilient crop yields (Chahal and Van Eerd, 2023). Soil structure improvement is one of the multiple services that CC fulfil in agroecosystems and as such part of a multifunctional concept of

CC. Specific CC mixtures could be designed to stimulate individual or multiple functions.

**Code and data availability**

All metadata and R code to reproduce the content of this study are publicly available under the Creative Commons Attribution 3.0 Germany. The files are accessible from the Zenodo achieve by the following DOI: 10.5281/zenodo.7147566 (https://doi.org/10.5281/zenodo.7147566).

**Author contributions**

NG designed the experiment, carried out the fieldwork and all statistical analyses. FLR participated in field work carried out most of the laboratory work and provided raw data. NG and FLR were writing the manuscript. DH was participating data analyses and helped with discussion and manuscript improvement. DS and UF maintained the field sited and carried out all agronomic measures. GG and JB developed the project ideas, recruited funding and participated in manuscript improvement.

**Competing interests**

The authors declare that they have no conflicts of interest.

**Acknowledgements**

This work is part of the BonaRes (Soil as a Sustainable Resource for the Bioeconomy) project CATCHY (Catch-cropping as an agrarian tool for continuing soil health and yield increase) funded by the German Federal Ministry of Education and

Research (BMBF), project number 031A559C. We are grateful to Silke Bokeloh and the whole laboratory team from the Institute of Soil Science for assistance with sample preparation and measurements.



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
