# Peer review of "Cover crops improve soil structure and change organic carbon distribution in macroaggregate fractions"

_EGUsphere, 2023_

## Referee Comment (RC1)

**General comments:**

This study explored the legacy effect of cover crops, including single and mixtures on soil aggregates in the second main crop rotation following cover crop treatments based on a long-term field experiment. The results of this study have certain meaningful for understanding the legacy effect of soil structure improvement for the subsequent main crops after consecutive cover cropping. However, there are still some issues that need to be addressed including statistical methods of significant difference, accuracy of result descriptions and partially redundant conclusions. Detailed comments are as below.

**Main comments:**

1. Why was the pairwise t-tests chosen instead of analysis of variance (ANOVA) to compare the significant difference between the seven treatments, e.g., Fig. S5?

2. It is very confusing to understand the results of significant difference between all indexes because the description in main text did not always show the consistency with figures, e.g., Fig. 2, Fig. S4, S5, S6, S7. For example, 'The significantly higher MWD was observed for clover at 0-10 cm (18.8% higher), Mix12 at 20-30 cm (37.6% higher) compared to the fallow in Line 200-201 (Fig. 2)'. However, lowercase letters of significant difference are 'b' for clover and 'a' for Mix12 in Fig. 2a. Please check and keep consistency throughout the manuscript.

**Specific comments:**

1. It would be better to add 1-2 sentences in Abstract to indicate the OC distribution within macroaggregate fractions.

2. Please add the details of planting date and harvest date for all crops in the section of Materials and Methods. For CC mixture treatment could add the ranges of planting and harvest date for cover crops.

3. 'OC2_1' is not defined in Line 163-164. Please check and add the missing information.

4. Please add the statistical analysis of significant difference and the range of *p* value in the section of Materials and Methods.

5. Please add subheadings for each independent results in the section of Results, e.g., 3.1 SOC concentrations and stocks. 3.2 Soil aggregates distribution….

6. Line 215. Please check the figure caption in Fig. S10. 'MWD' change as 'GMD'?

7. The first paragraph in Discussion section is overlap with the introduction, objectives of the study and materials. Please rewrite.

8. Line 344-350. Please refined and delete the citation. It would be better if the authors could add 1 or 2 statements to expand the future research recommendations.

9. The document of supplementary material S2 is missing. Please check.

**Figures.**

1. Fig. 2. Please delete 'Lowercase letters denote' before '(a)… and (b)…' in figure caption.

2. There is no citation for Fig. S11 in the main text.

---

## Author Comment (AC1)

**Response to Reviewer #1**

We thank the reviewer for the constructive discussion. We took all the comments very seriously and improved the manuscript further as suggested. In the following the comments of the reviewers are numbered and in cursive followed by our response in plain. Changes in the manuscript can be followed in the revision mode. A new R script was compiled and uploaded to the server.

**Main comments:**

1. *Why was the pairwise t-tests chosen instead of analysis of variance (ANOVA) to compare the significant difference between the seven treatments, e.g., Fig. S5?*

There are, of course, various different methods to evaluate differences between treatments. Pairwise t-tests and ANOVA followed by a post hoc test (usually Tukey's HSD) are the most commonly used and there is no clear rule what to use in which situation. Usually both methods can be used in exchange, but there are some subtle differences. First, F statistic of an ANOVA is not always a robust test. Particularly if the variances across the groups are not equal. Thus, the assumptions that F-statistic is reliable is the assumption that the variances of the groups are equal. Let's focus for example on the following example below (Fig.2a upper facet 0-10 cm):

[Figure]

We see by eye that the variances of the treatments are very different to each other. This is one of the reasons why we show all the data in the plot, not just mean and SE. We can run a Leven's test to prove that:

```
Levene's Test for Homogeneity of Variance (center = median)
      Df F value Pr(>F)
group  6  0.5667 0.7502
      14
```

As you can see the assumption of homogeneity of variances is violated because $Pr(>F)$ is larger 0.05, and F value is below 1. So, we cannot run an ANOVA nor a Student's t-Test (same assumption as ANOVA). The only option is a Welch's t-Test which does not have the assumption of homogeneity of variances. This is what we did with the *paiwiseTest* function from the *pairwiseCI* R package.

```
P-values calculated using
 Welch Two Sample t-test

              p.value
```

```
Mustard-Fallow      0.8797
Clover-Fallow       0.0173
Oat-Fallow          0.7949
Phacelia-Fallow     0.7769
Mix4-Fallow         0.4554
Mix12-Fallow        0.1238
Clover-Mustard      0.0268
Oat-Mustard         0.9607
Phacelia-Mustard    0.8378
Mix4-Mustard        0.5228
Mix12-Mustard       0.1398
Oat-Clover          0.0054
Phacelia-Clover     0.2294
Mix4-Clover         0.1718
Mix12-Clover        0.7002
Phacelia-Oat        0.8463
Mix4-Oat            0.5066
Mix12-Oat           0.1531
Mix4-Phacelia       0.8037
Mix12-Phacelia      0.3341
Mix12-Mix4          0.3482
```

Another option would be a relatively new approach of estimation plots introduced by (Ho et al., 2019). Estimation plots have the advantage not to depend on p-values and null-hypothesis significance testing. According to this method the example from above would look like the following:

[Figure]

According to the estimation plots not only clover would be different from the fallow (as the Welch test above showed) but also Mix12. But we have refrained from using this kind of statistic across all different comparisons. It would be too complex and overloading in the Manuscript.

2. *It is very confusing to understand the results of significant difference between all indexes because the description in main text did not always show the consistency with figures, e.g., Fig. 2, Fig. S4, S5, S6, S7. For example, 'The significantly higher MWD was observed for*

*clover at 0-10 cm (18.8% higher), Mix12 at 20-30 cm (37.6% higher) compared to the fallow in Line 200-201 (Fig. 2)'. However, lowercase letters of significant difference are 'b' for clover and 'a' for Mix12 in Fig. 2a.Please check and keep consistency throughout the manuscript.*

Thank you for the hint. Indeed, it appeared to be a bit confusing. Note, the pairwise comparison was done for each soil horizon separately. So, it does not matter for the scientific correctness how the order of the levels is. I marked that in the manuscript as well. Nevertheless, I wrote a new function that reordered the levels of the data so that the letters appear in an ascending order of the mean. From now on, the letter "a" starts always with the treatment with the lowest mean. It is, however, not possible to order the levels in a in a specific fixed order of the treatments. This would be possible only manually during plotting. As we wanted to have a machine-readable script, that could be accessed from everyone without explanation, we refrain from a manual sorting. The new function was applied to all figures that used pairwise t-tests. An updated R script was uploaded to the Zenodo server.

**Specific comments:**
*1. It would be better to add 1-2 sentences in Abstract to indicate the OC distribution within macroaggregate fractions.*

We added 2 sentences.

*2. Please add the details of planting date and harvest date for all crops in the section of Materials and Methods. For CC mixture treatment could add the ranges of planting and harvest date for cover crops.*

The information were added as supplementary Table S3.

*3. 'OC2_1' is not defined in Line 163-164. Please check and add the missing information.*

It is explained in the next sentences: "Principal component analyses confirmed a similar loading of OC2_1 and OC4_2 on the first two components (eigenvalue >1), explaining 61% of the variance in the data. As OC2_1 and OC4_2 are redundant variables, including both does not fit to the model structure. Thus, we excluded OC2_1 from the latent variable construction."

*4. Please add the statistical analysis of significant difference and the range of p value in the section of Materials and Methods.*

Done.

*5. Please add subheadings for each independent results in the section of Results, e.g., 3.1 SOC concentrations and stocks. 3.2 Soil aggregates distribution….*

Subheadings were included.

*6. Line 215. Please check the figure caption in Fig. S10. 'MWD' change as 'GMD'?*

No, everything is correct. Maybe it was a bit confusing because we finished the last paragraph the GMD.

*7. The first paragraph in Discussion section is overlap with the introduction, objectives of the study and materials. Please rewrite.*

Yes, I always like to recall the basic aims of the study before starting the discussion. But I agree that the text is redundant an removed it.

*8. Line 344-350. Please refined and delete the citation. It would be better if the authors could add 1or 2 statements to expand the future research recommendations.*
Done.

*9. The document of supplementary material S2 is missing. Please check.*

Right. S2 is the R markdown file. I wanted to include this file after the review when the script is finalized.

**Figures.**
*1. Fig. 2. Please delete 'Lowercase letters denote' before '(a)… and (b)…' in figure caption.*

We changed the description

*2. There is no citation for Fig. S11 in the main text.*

There was a mistake in line 225. I wrote Fig. S10 instead S11. It is corrected now.

**References**:

Ho, J., Tumkaya, T., Aryal, S., Choi, H., and Claridge-Chang, A.: Moving beyond P values: data analysis with estimation graphics, Nat Methods, 16, 565–566, https://doi.org/10.1038/s41592-019-0470-3, 2019.

---

## Author Comment (AC2)

**Response to Reviewer #2**

We thank the reviewer for the constructive discussion. We took all the comments very seriously and improved the manuscript further as suggested. In the following the comments of the reviewers are numbered and in cursive followed by our response in plain. Changes in the manuscript can be followed in the revision mode. A new R script was compiled and uploaded to the server. Also S2, the R markdown file was finalized.

*The manuscript explores the changes in soil structure and carbon content through different cover crops, it is important for further increasing soil carbon content and improving soil structure. However, this manuscript needs be further revised and improved as shown the following issues:*

1. *Although different CCs were selected in this manuscript, what is the basis for selecting these CCs?*

We selected cover crops that reflect several criterion. (1) the CC should be used widely in the practical application. Mustard is one of the cheapest and most widely used CC by farmers in Europe. The same is true for phacelia that is not related to any plant family of cash crops. (2) the plants should represent contrasting plant families with different properties and functional traits. Mustard: *Brassikaceae* with high production of allelopathic substances (like Gucosinolates) also sometimes used to biofumigation approaches. Deep, tab roots until ~1m, high N uptake. Phacelia: *Boraginaceae*, is widely used before sugar beets to reduce nematode pressure. Contained high capacity for P mobilisation from soil sources and high P concentrations in biomass. Bristl oat: *Poaceae*, brings a high rooting density and volume in the upper 50 cm and high N uptake. Egyptian clover: *Fabaceae*, N uptake from atmosphere, low C/N ratio in roots and shoots. Shallow rooting system but high input of rhizosphere products. Mix4 was a mixture of all species at a ratio of 25% each by seeds. Mix12 should represent a maximum of biodiversity with all plant families as above plus some more. The mixture should contain plants with different functional traits to explore niches in soil and nutrients. Further, all CC can be frost killed and do not require herbicide application in regions with cold winters.

2. *The role of different CCs was involved in the discussion, but this article does not provide the data of CC. Please supplement these data (such as root morphology, CC carbon type, root density, etc.) to support the discussion;*

Plant data, root morphology and root exudation patter are presented in the following publications that were cited in the manuscript:

Gentsch, N., Boy, J., Batalla, J. D. K., Heuermann, D., von Wirén, N., Schweneker, D., Feuerstein, U., Groß, J., Bauer, B., Reinhold-Hurek, B., Hurek, T., Céspedes, F. C., and Guggenberger, G.: Catch crop diversity increases rhizosphere carbon input and soil microbial biomass, Biol Fertil Soils, https://doi.org/10.1007/s00374-020-01475-8, 2020.

Gentsch, N., Heuermann, D., Boy, J., Schierding, S., von Wirén, N., Schweneker, D., Feuerstein, U., Kümmerer, R., Bauer, B., and Guggenberger, G.: Soil nitrogen and water management by winter-killed catch crops, SOIL, 8, 269–281, https://doi.org/10.5194/soil-8-269-2022, 2022.

Heuermann, D., Gentsch, N., Boy, J., Schweneker, D., Feuerstein, U., Groß, J., Bauer, B., Guggenberger, G., and Wirén, N. von: Interspecific competition among catch crops modifies vertical root biomass distribution and nitrate scavenging in soils, Sci Rep, 9, 1–11, https://doi.org/10.1038/s41598-019-48060-0, 2019.

Heuermann, D., Gentsch, N., Guggenberger, G., Reinhold-Hurek, B., Schweneker, D., Feuerstein, U., Heuermann, M. C., Groß, J., Kümmerer, R., Bauer, B., and von Wirén, N.: Catch crop mixtures have higher potential for nutrient carry-over than pure stands under changing environments, European Journal of Agronomy, 136, 126504, https://doi.org/10.1016/j.eja.2022.126504, 2022.

Heuermann, D., Döll, S., Schweneker, D., Feuerstein, U., Gentsch, N., and von Wirén, N.: Distinct metabolite classes in root exudates are indicative for field- or hydroponically-grown cover crops, Frontiers in Plant Science, 14, 2023.

All publication produced data from the same set of CC at the same long-term experiment. Therefore, we like to refer to these publications instead of presenting to much details here. We added a paragraph in the material and method section. We also added Table S3 in the supplement on shoot-root biomass and C:N ratios.

3. *Please provide field management in the material method;*

Management is explained in detail in the supplement material S1. But we add a sentence more.

4. *Added P values after the significant results, and delete the non-significant results to further refine the results*

We added the following sentence: "The cut of between the terms "significant" and "not significant" was defined to $p > 0.05$." Further p-values were added where needed in the results section and non significant p-values were deleted.

5. *The manuscript mentions the contribution of fungi to soil structure, such as L73, L281, etc. However, the number of fungi decreases and the number of bacteria increases due to fertilization in farmland. Moreover, there is no relevant fungal data in this manuscript. Please further revise and improve;*

That is right. Farmland has very low Fungi:Bacteria ratios compared to grassland or native vegetation. But it depends on the management. Reduction of soil cultivation increases the Fungal activity and biomass (Helgason et al., 2010), the same for CC applications (Cloutier et al., 2020; Thapa et al., 2021; Gentsch et al., 2020). With respect to fungi we refer in the introduction and discussion to our study (Gentsch et al., 2020) from the same site (see figures below). We found that CC stimulate particularly the fungal activity in the experiments. Fungi:Bacteria ratios increase and also the Fungal Biomass increased during CC growth. The data below based on PLFA measurements during CC growth in October. We found with a $^{13}$C labelling technique, that fresh photosynthesis C compounds from CC were transported mainly to fungi. We add some more references in the discussion.

[Figure]

6. *The figures in the manuscript provide the replicates data, such as Figure 2, and the figures in the supporting information. It is enough to the mean with standard error in the figures. Please further modify and add statistical analysis in the figures;*

Unfortunately, we disagree with this suggestion. Just to show means and standard errors is not enough to understand the complexity of the data and the selection of the statistical methods. As explained in the answer of reviewer #1, it is possible so explore the variance and the distribution of the original data and justify the decision to use e.g. Welch's pairwise t-Test. We like to refer to the publication of (Ho et al., 2019), who clearly explained the advantages of data transparency and moving beyond p-values.

7. *L252-255 is the result, please move to the results parts; Further refine the discussion section;*

The results were already described in the results part in detail. Here we need to call back the reader's attention to the results that we discuss below. We like to refer to the famous book of Schimel (2012), who stated that repeating the most important results is an integral part story telling in the discussion section. These principles are also stated in the academic phrase bank of Manchester University (https://www.phrasebank.manchester.ac.uk/discussing-findings/). Restarting the results and cantering the discussion around these results is one of the keys of a successful discussion. Each of our discussion paragraphs is following this rule.

8. *L292 nuclear magnetic resonance data, not included in the manuscript, please delete it;*

Off course we do not show NMR data. We also discuss a lot of microbial related functions and do not show microbial data. Most of the literature we cite used methods that we do not use here. The sense of this section is to discuss data, that relate to our CC study and might explain changes of MWD due to plant inputs. We like to draw the discussion on the importance of OM chemistry for the MWD and aggregation of soil particles. OM chemistry in terms of litter quality is an important factor that that is one of the 4 controls we outlined in the introduction (L54 following). We like to consider all the 4 aspects in the discussion. Without discussion of carbon quality parameters, the discussion would be incomplete. Unless litter C:N ratios and other element stoichiometry we do not have insights to litter quality parameters. Unfortunately, there are not so many new studies on NMR and CC litter quality. But we added some more references to support the discussion (Jensen et al., 2005; Husain and Dijkstra, 2023; Halder et al., 2021; Oliveira et al., 2016).

*9.  L294 mentions cellulose hemicellulose, please provide relevant data for CC;*

We provide several references.

*10. Is "the strongest direct effect" in L311 significant?*

Yes, shown in Fig.4, numbers show standardized estimates with p values as asterisks.

*11. L320 not clear;*

Right, it was just a hypothesis that we could not support with data. So, we delete the sentence.

*12. Two largest fractions, which fractions?*

The 4-8 and 8-16 mm fraction, mentioned in the sentence before. I guess it is clear that the sentences are connected together.

*13. Rhizosphere products and hyphae, providing data support*

The last paragraph is just a short synopsis of the discussion. Rhizosphere products and fungal impacts are discussed in L285 to 306 with many references. The statement in L352 refers to the reference Tisdall and Oades (1982).

*14. Please change the conclusion to a paragraph*

Done.

**References:**

Cloutier, M., Murrell, E., Barbercheck, M., Kaye, J., Finney, D., García-González, I., and Bruns, M.: Fungal community shifts in soils with varied cover crop treatments and edaphic properties, Scientific Reports, 10, https://doi.org/10.1038/s41598-020-63173-7, 2020.

Gentsch, N., Boy, J., Batalla, J. D. K., Heuermann, D., von Wirén, N., Schweneker, D., Feuerstein, U., Groß, J., Bauer, B., Reinhold-Hurek, B., Hurek, T., Céspedes, F. C., and Guggenberger, G.: Catch crop diversity increases rhizosphere carbon input and soil microbial biomass, Biol Fertil Soils, https://doi.org/10.1007/s00374-020-01475-8, 2020.

Halder, M., Liu, S., Zhang, Z. B., Guo, Z. C., and Peng, X. H.: Effects of residue stoichiometric, biochemical and C functional features on soil aggregation during decomposition of eleven organic residues, CATENA, 202, 105288, https://doi.org/10.1016/j.catena.2021.105288, 2021.

Helgason, B. L., Walley, F. L., and Germida, J. J.: No-till soil management increases microbial biomass and alters community profiles in soil aggregates, Appl. Soil Ecol., 46, 390–397, https://doi.org/10.1016/j.apsoil.2010.10.002, 2010.

Ho, J., Tumkaya, T., Aryal, S., Choi, H., and Claridge-Chang, A.: Moving beyond P values: data analysis with estimation graphics, Nat Methods, 16, 565–566, https://doi.org/10.1038/s41592-019-0470-3, 2019.

Husain, H. and Dijkstra, F. A.: The influence of plant residues on soil aggregation and carbon content: A meta-analysis, Journal of Plant Nutrition and Soil Science, 186, 177–187, https://doi.org/10.1002/jpln.202200297, 2023.

Jensen, L. S., Salo, T., Palmason, F., Breland, T. A., Henriksen, T. M., Stenberg, B., Pedersen, A., Lundström, C., and Esala, M.: Influence of biochemical quality on C and N mineralisation from a broad variety of plant materials in soil, Plant Soil, 273, 307–326, https://doi.org/10.1007/s11104-004-8128-y, 2005.

Oliveira, R. A. de, Brunetto, G., Loss, A., Gatiboni, L. C., Kürtz, C., Müller Júnior, V., Lovato, P. E., Oliveira, B. S., Souza, M., and Comin, J. J.: Cover Crops Effects on Soil Chemical Properties and Onion Yield, Rev. Bras. Ciênc. Solo, 40, e0150099, https://doi.org/10.1590/18069657rbcs20150099, 2016.

Schimel, J.: Writing science: how to write papers that get cited and proposals that get funded, OUP USA, 2012.

Thapa, V. R., Ghimire, R., Acosta-Martínez, V., Marsalis, M. A., and Schipanski, M. E.: Cover crop biomass and species composition affect soil microbial community structure and enzyme activities in semiarid cropping systems, Applied Soil Ecology, 157, 103735, https://doi.org/10.1016/j.apsoil.2020.103735, 2021.

---

## Author Response (AR2)

**Response to the Editor**

We thank the editor for the thorough review. Of course, the correct formulation must be $p < 0.05$.
We apologize for the typo and will correct the manuscript accordingly.